# A bacterial small RNA regulates the adaptation of *Helicobacter pylori* to the host environment

Ryo Kinoshita-Daitoku[1,2], Kotaro Kiga[2], Masatoshi Miyakoshi [3], Ryota Otsubo[1,12], Yoshitoshi Ogura[4,5], Takahito Sanada[1,2], Zhu Bo[2], Tuan Vo Phuoc[6,7], Tokuju Okano[8], Tamako Iida[2], Rui Yokomori[9], Eisuke Kuroda[1,2], Sayaka Hirukawa[2], Mototsugu Tanaka[2,8,10], Arpana Sood[2], Phawinee Subsomwong[1], Hiroshi Ashida[8], Tran Thanh Binh[6,7], Lam Tung Nguyen[6], Khien Vu Van[11], Dang Quy Dung Ho[7], Kenta Nakai [9], Toshihiko Suzuki[8], Yoshio Yamaoka [6], Tetsuya Hayashi[4] & Hitomi Mimuro [1,2✉]

Long-term infection of the stomach with *Helicobacter pylori* can cause gastric cancer. However, the mechanisms by which the bacteria adapt to the stomach environment are poorly understood. Here, we show that a small non-coding RNA of *H. pylori* (HPnc4160, also known as IsoB or NikS) regulates the pathogen's adaptation to the host environment as well as bacterial oncoprotein production. In a rodent model of *H. pylori* infection, the genomes of bacteria isolated from the stomach possess an increased number of T-repeats upstream of the HPnc4160-coding region, and this leads to reduced HPnc4160 expression. We use RNA-seq and iTRAQ analyses to identify eight targets of HPnc4160, including genes encoding outer membrane proteins and oncoprotein CagA. Mutant strains with HPnc4160 deficiency display increased colonization ability of the mouse stomach, in comparison with the wild-type strain. Furthermore, HPnc4160 expression is lower in clinical isolates from gastric cancer patients than in isolates derived from non-cancer patients, while the expression of HPnc4160's targets is higher in the isolates from gastric cancer patients. Therefore, the small RNA HPnc4160 regulates *H. pylori* adaptation to the host environment and, potentially, gastric carcinogenesis.

[1] Department of Infection Microbiology, Research Institute for Microbial Diseases, Osaka University, Osaka, Japan. [2] Division of Bacteriology, Department of Infectious Diseases Control, International Research Center for Infectious Diseases, The Institute of Medical Science, The University of Tokyo, Tokyo, Japan. [3] Division of Biomedical Science, Faculty of Medicine, University of Tsukuba, Tsukuba, Japan. [4] Department of Bacteriology, Graduate School of Medical Sciences, Kyushu University, Fukuoka, Japan. [5] Division of Microbiology, Department of Infectious Medicine, Kurume University School of Medicine, Kurume Fukuoka, Japan. [6] Department of Environmental and Preventive Medicine, Faculty of Medicine, Oita University, Yufu Oita, Japan. [7] Department of Endoscopy, Cho Ray Hospital, Ho Chi Minh, Vietnam. [8] Department of Bacterial Pathogenesis, Infection and Host Response, Graduate School of Medical and Dental Sciences, Tokyo Medical and Dental University, Tokyo, Japan. [9] Human Genome Center, The Institute of Medical Science, The University of Tokyo, Tokyo, Japan. [10] Division of Nephrology and Endocrinology, The University of Tokyo School of Medicine, Tokyo, Japan. [11] Department of GI Endoscopy, 108 Central Hospital, Hanoi, Vietnam. [12] Present address: Laboratory of Pharmaceutical Integrated Omics, Department of Pharmaceutical Engineering, Facility of Engineering, Toyama Prefectural University, Toyama, Japan. ✉email: mimuro@biken.osaka-u.ac.jp

The gastric pathogen *Helicobacter pylori* infects approximately half of the world's population and increases the risk of developing peptic ulcers, chronic gastritis, intestinal metaplasia, and gastric cancer[1–3]. The genomic diversity characteristic of *H. pylori* is crucial for its establishment of persistent infections in hosts and confers its adaptability to extreme gastric environments[4,5]. A well-known example of *H. pylori*'s strong diversity is the presence of outer membrane proteins (OMPs), which are highly antigenic cell surface proteins and are thought to allow *H. pylori* to escape from host immunity during chronic infection[6]. Mutations in OMP genes are characterized by simple repetitive sequences such as mononucleotide repeats (e.g., poly-T) and dinucleotide repeats (e.g., CT-repeats)[6]. The expansion and contraction of these simple repetitive sequences result in a genetically heterogeneous bacterial population, with phase variations controlled by the ON/OFF gene expression of the proteins[4,6–8]. However, whether the *H. pylori* adaptation to the host is controlled only by the diversity of mRNA containing simple repetitive sequences remains unclear. Because of the prevalence of small regulatory RNAs (sRNAs) in *H. pylori*[9], it is also possible that a heterogeneously mutated sRNA changes the expression levels of its target mRNAs or a mutation in the target sequence results in dysregulation by an sRNA[10]. Therefore, to understand the adaptation mechanisms of *H. pylori*, we analyzed bacterial gene mutations acquired by *H. pylori* over the course of infection using an experimental animal infection system in hosts of identical genetic background.

Here, we show that a small non-coding RNA HPnc4160 of *H. pylori* regulates the pathogen's adaptation to the host environment as well as bacterial oncoprotein production. We revealed that during the infection, a T-repeat length upstream of the coding region for HPnc4160 is elongated in *H. pylori* and that this elongation leads to decreased expression of HPnc4160. Through RNA-seq and iTRAQ analysis we identified several targets of HPnc4160, including *cagA*, a known carcinogenic protein. Importantly, we found that HPnc4160 expression is lower in clinical isolates from gastric cancer patients than in isolates derived from non-cancer patients, while the expression of HPnc4160's targets is higher in the isolates from gastric cancer patients. Our findings demonstrate that the small RNA HPnc4160 regulates *H. pylori* adaptation to the host environment and, potentially, gastric carcinogenesis.

## Results

**Identification of mutations accumulated during infection.** To analyze the bacterial gene mutations acquired by *H. pylori* during persistent infection, Mongolian gerbils ($n = 10$) were inoculated with the *H. pylori* ATCC 43504 wild-type strain for 8 weeks with stable gastric colonization[11]. *H. pylori* were isolated from the infected stomachs (4 clones per gerbil, total 40 clones; Fig. 1a) and then analyzed by comparative whole-genome sequencing (Supplementary Fig. 1a, Supplementary Data 1 and 2). By integrating the genomic positions of these mutations, we identified 13 regions (R1, R3–R5, R7–R8, R10–R16) in which mutations were introduced in 50% or more of the strains (Supplementary Fig. 1a, Supplementary Data 2).

To investigate whether these mutated regions affect gene expression, transcripts in the mutated regions were quantified in isolates obtained from gerbils. Among the 15 corresponding coding (CDS) and non-coding RNA sequences (HP0947, *babA*, *tpiA*, jhp1163, HP0811, HPnc4160, HPnc4170, jhp0540, *araS*, *pldA*, *sabA*, HP1354, *hopZ*, *tlpB*, and HPB8_818), the expression of HPnc4160 showed the greatest fluctuation (Fig. 1b, Supplementary Fig. 1a, b). Similar results were obtained from strains (2 clones per mouse, total 10 clones) isolated from C57BL/6 mice

($n = 5$) infected with *H. pylori* ATCC 43504 wild-type strain for 8 weeks (Supplementary Fig. 1a–c; Supplementary Data 3 and 4).

HPnc4160 (IsoB) was previously proposed as a *cis*-acting sRNA for HPnc4170 (*aapB* small ORF homolog) (Supplementary Fig. 2a)[9,12]. HPnc4160 and its upstream T-repeat region are conserved in *H. pylori* strains, but the T-repeat length in the promoter region is highly variable (Supplementary Fig. 2b). One to four additional thymidine bases were inserted into the T-repeat of our isolates from rodents, with the repeat length increasing with duration of period (Fig. 1c, Supplementary Fig. 2c–f). However, expansion of this region was not observed in long-term in vitro culture of wild-type strain (Supplementary Fig. 2g). Next, we analyzed changes in HPnc4160 expression according to the T-repeat length. In strains recovered from the *H. pylori*-infected stomachs of gerbils, HPnc4160 expression decreased with the expansion of the T-repeat (Fig. 1d, e). To exclude the effects of mutations other than those of T-repeats in strains recovered from rodents, we further analyzed the RNA expression of HP0811, HPnc4170, and HPnc4160 in isogenic variants of ATCC 43504 wild-type strains harboring different T-repeat lengths at the promoter region of HPnc4160 (T0 to T20, Fig. 1f, g, Supplementary Fig. 3a–c). The expression of HPnc4160 with varying T-repeat lengths was multiphasic: low in T1, high in T4, high in T14 (number of repeats in ATCC 43504 wild-type), low in T16, and intermediate in T18 (Fig. 1f, g). Expression levels of *hpnc4170* and its downstream HP0811 were not affected by the T-repeats (Supplementary Fig. 3b–c), indicating that HPnc4160 is unlikely to act as a *cis*-acting sRNA.

**Identification of HPnc4160 target genes.** *Helicobacter pylori* utilizes RNA-mediated regulation *in trans* through relatively short base-pairing with multiple mRNAs from different loci despite the absence of known RNA matchmakers such as Hfq and ProQ[13,14]. To identify the target mRNA of HPnc4160, we generated a Δ*hpnc4160-hpnc4170* strain, in which both HPnc4160 and HPnc4170 on the complementary strand were deleted. mRNA and protein expression analysis identified eight differentially expressed factors in the mutant strain compared to in the wild-type (*cagA*, *hofC*, HELPY_1262, HP0410, *horB*, *omp14*, *hopE*, and HP1227; $P < 0.001$ by RNA-seq; $P < 0.01$ by isobaric tag for relative and absolute quantitation labeling and liquid chromatography-tandem mass spectrometry analysis) (Fig. 2a–c, Supplementary Data 5 and 6). Notably, five of these eight factors (HofC, HP0410, HorB, Omp14, and HopE) were annotated as outer membrane proteins. More strikingly, the expression of *cagA* mRNA and protein, which is a known bacterial oncoprotein[1], were most noticeably increased (Fig. 2a, b, Supplementary Data 5 and 6).

We analyzed whether the mRNA expression of these eight factors depended on the presence of HPnc4160. Levels of HPnc4160 trended to decrease, with T16 exhibiting the lowest value, whereas the mRNA of the eight candidates trended to increase, with T16 exhibiting the highest value (Fig. 1f, g; Supplementary Fig. 3a–d). The expression of these target mRNAs and the presence of HPnc4160 showed a strong inverse correlation (Spearman correlation coefficient ($r$) = −0.8234 to −0.7312) (Supplementary Fig. 3d). The HPnc4160 overexpression strain (WT/pHel2-*hpnc4160*) showed significantly decreased expression of the target mRNAs. In contrast, in the Δ*hpnc4160-hpnc4170* strain, the mRNA expression of each target increased compared to in the wild-type (Supplementary Fig. 3e–h). As the Δ*hpnc4160-hpnc4170* strain lacks the HPnc4170 sequence in the complementary strand of HPnc4160, we constructed a Δ*hpnc4160-hpnc4170*/pHel2-*hpnc4160* strain complementing only HPnc4160 to confirm the effect of the HPnc4170 sequence on HPnc4160 target mRNA expression. Compared to the

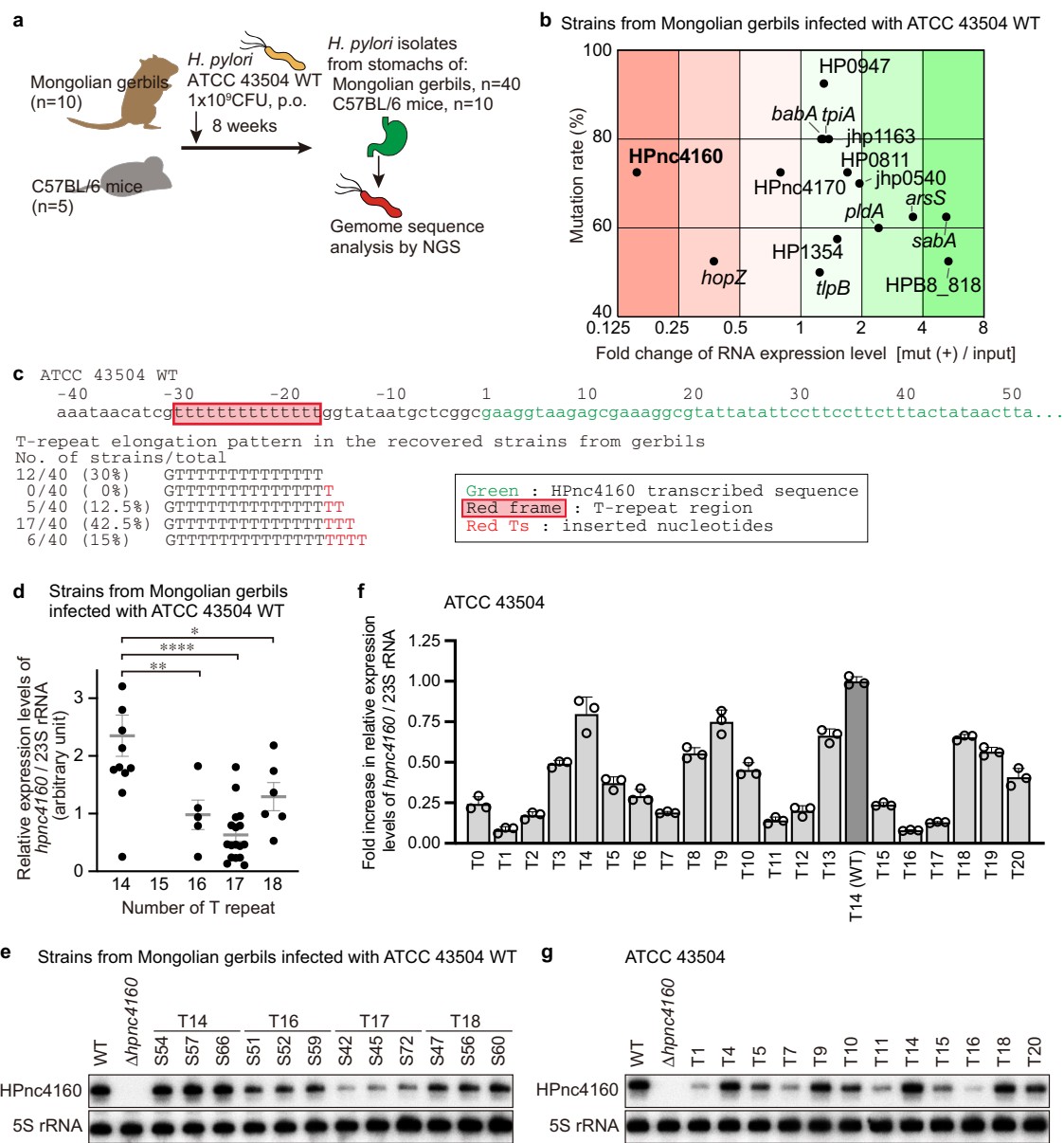

**Fig. 1 H. pylori acquires T-repeat extension upstream of HPnc4160 to decrease its expression during infection in vivo. a** Schematic showing experimental strategy. **b** RNA expression levels of ORFs or nearby genes of genome regions mutated in >50% of the strains recovered from gerbils, plotted against mutation rates. **c** DNA sequence around the HPnc4160 and T-repeat sequence of the strains recovered from gerbils. **d–g** Relative *hpnc4160* expression in *H. pylori* strains recovered from gerbils (T repeat number of 14, *n* = 12; 16, *n* = 5; 17, *n* = 17; 18, *n* = 6) and genetically modified T-repeat lengths (*n* = 3) in RT-PCR (**d**, **f**) and Northern blot (**e**, **g**). Shown are the means ± s.d (**d**, **f**). Data were processed by Grubbs' test for outliers. *P = 0.0379, **P = 0.0088, ****P < 0.0001 by Sidak's multiple comparison tests (two sided) (**d**). Data are representative of two independent experiments (**d–g**). Source data are provided as a Source Data file.

Δ*hpnc4160-hpnc4170* strain, the mRNA expression levels of the candidates were decreased in the HPnc4160 complemented Δ*hpnc4160-hpnc4170* /pHel2-*hpnc4160* strain (Supplementary Fig. 3h). These data indicate that the eight target candidates are negatively regulated by HPnc4160.

*Trans*-acting sRNAs generally form base-pairing in the 5′ untranslated region (UTR) or CDS of target mRNAs to repress or activate gene expression at the post-transcriptional level[15,16]. In *H. pylori*, the best-characterized sRNA RepG (HPnc5490) binds to G-repeats in 5′UTR of *tlpB* mRNA to repress translation and destabilize mRNA[9,10]. As *H. pylori* lacks RNase E/G homologs[13,17], which promote mRNA degradation by sRNAs, other endoribonuclease such as RNase III may degrade target mRNA accompanied by translation inhibition.

In the 5′UTR of seven identified target genes except *cagA*, we found a sequence that binds complementarily to the loop region of the HPnc4160. (Supplementary Fig. 4a, b). The direct binding of HPnc4160 to the 5′UTR of the seven target genes except *cagA* were confirmed by Electrophoretic mobility shift assays (Fig. 3a). Instead, we identified five putative target sites in the CDS of *cagA* (Type 1 at 2344 nt from the start codon, and Type 2 at four positions of 2838, 2940, 3042, and 3144 nt) (Fig. 3b–d). We confirmed the direct binding of partial *cagA* CDS (positions 2778–3236 nt from start codon of *cagA*) to HPnc4160 (Fig. 3e). This binding was abolished for Non-Binding (NB)-*cagA* RNA in which the four Type 2 HPnc4160 binding sequences were mutated and the amino acid sequence of CagA was preserved (Fig. 3b, e–g). The abolished binding of NB-*cagA* RNA was

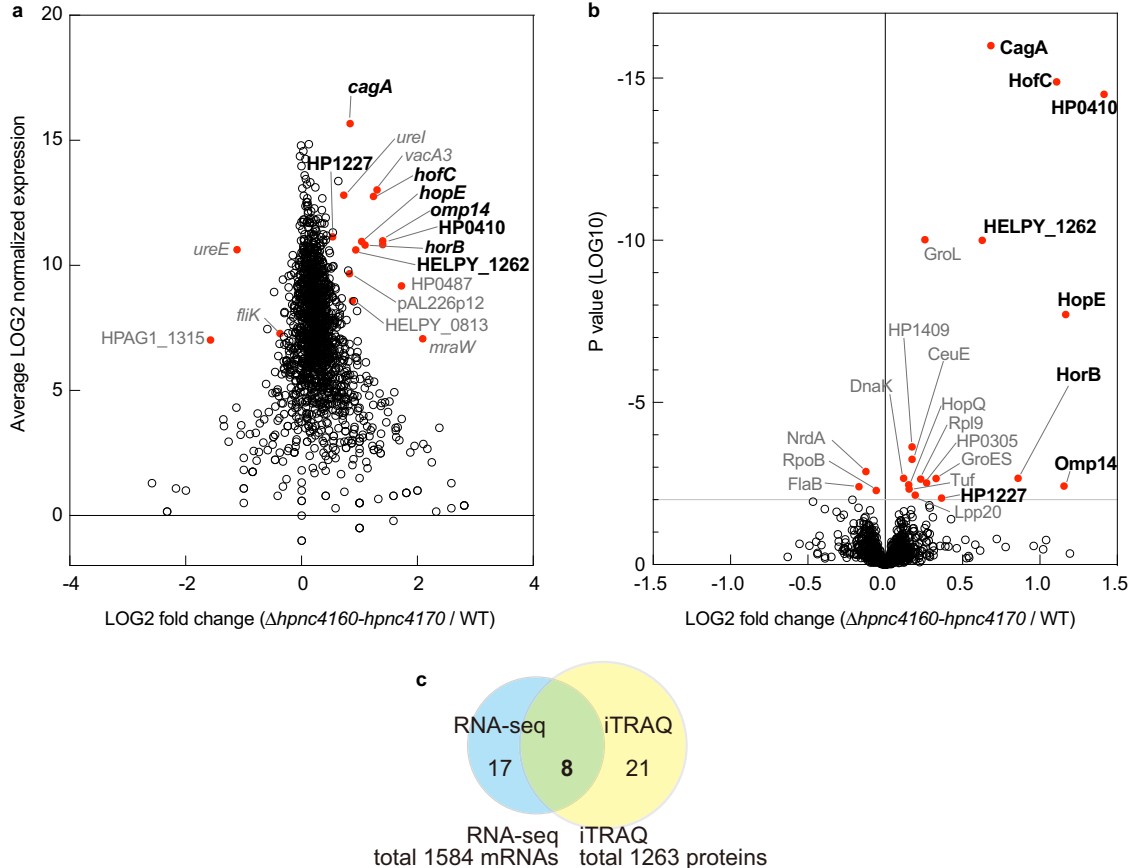

**Fig. 2 HPnc4160 downregulates the expression of bacterial pathogenic factors. a** MA plot of ratios [Δ*hpnc4160-hpnc4170* / wild-type (WT) *H. pylori*] versus their normalized average mRNA expression determined by RNA-sequencing (RNA-seq). Red dots, *P* < 0.001 by a Fisher Exact Test (two-sided). **b** Volcano plot of proteins quantified by isobaric tags for relative and absolute quantification analysis comparing WT and Δ*hpnc4160-hpnc4170*. Each point represents the difference in protein expression (fold-change) between the two groups plotted against the level of statistical significance. Red dots, *P* < 0.01 by the paired t-test (two-sided). **c** Venn diagram representing the number of significantly differentially expressed factors between Δ*hpnc4160-hpnc4170* and WT. Source data are provided as a Source Data file.

restored by complementary mutations in HPnc4160 at positions 64, 67, and 73 (HPnc4160 mut; Fig. 3e, h), demonstrating direct base-pairing between HPnc4160 and *cagA*. In addition, we found that *H. pylori* RNase III recombinant protein degraded the dsRNA upon binding between HPnc4160 and biotin-labeled partial *cagA* mRNA, but not NB-*cagA* RNA (Fig. 3i, Supplementary Fig. 4c). These data indicate that HPnc4160 controls *cagA* at the post-transcriptional level by binding to multiple binding sequences present in its CDS region and possibly promoting the degradation of mRNA by RNase III.

**Effects of HPnc4160 on *H. pylori* pathogenicity.** We further analyzed CagA, which has been shown to be deeply involved in pathogenesis[1–3]. The *H. pylori* expressing NB-*cagA*, in which all five HPnc4160-binding sequences were mutated but the amino acid sequence was preserved, showed significantly increased expression of *cagA* mRNA and protein to the same extent as that of the Δ*hpnc4160-hpnc4170* strain (Supplementary Fig. 5a–c). The increased expression of *cagA* mRNA and protein was negated by complementary mutations in *hpnc4160* (*hpnc4160* mut) (Supplementary Fig. 5a–c), indicating that HPnc4160 negatively controls the expression of *cagA* mRNA and protein in *H. pylori*. *H. pylori* injects CagA proteins into the host epithelium via a Type IV secretion system (TFSS), and then intracellular CagA proteins are phosphorylated at tyrosine residues (pY) by the host

Src/Abl kinase[18,19]. Using pY-CagA-specific antibodies, we confirmed that intracellular CagA increased in NB-*cagA*-infected cells, accompanied by an increase in total CagA, but not in *hpnc4160* mut strain-infected cells (Supplementary Fig. 5d). Intracellular CagA induces AGS cell motility (scattering/hummingbird phenotype)[20,21]. More elongated AGS cells were observed during infection with Δ*hpnc4160-hpnc4170* or NB-*cagA* strains vs in the wild-type or Δ*hpnc4160-hpnc4170*/pHel2-*hpnc4160* strain-infected cells (Supplementary Fig. 5e, f). The amount of IL-8 secreted from *H. pylori*-infected cells, which is primarily induced by intracellular CagA[22], was higher in NB-*cagA* strain-infected cells than in wild-type strain or *hpnc4160* mut strain-infected cells (Supplementary Fig. 5g). These results suggest that the binding of HPnc4160 to *cagA* mRNA is critical for controlling the amount of functional CagA protein in *H. pylori*.

We further assessed changes in CagA activity dependent on T-repeat length of *hpnc4160*. Western blot analysis showed that 48 out of 50 strains recovered from rodents infected with ATCC 43504 at 8 weeks post infection expressed VirB7 (TFSS protein) and retained the ability to translocate CagA into host epithelium (Supplementary Fig. 5h, i). Intracellularly translocated CagA protein tends to be higher in strains with T-repeat numbers 16 and 17 with low levels of HPnc4160, and lower in T14 and T18 with high levels of HPnc4160. A similar trend of increased IL-8 expression with T-repeat length was observed in the expression of

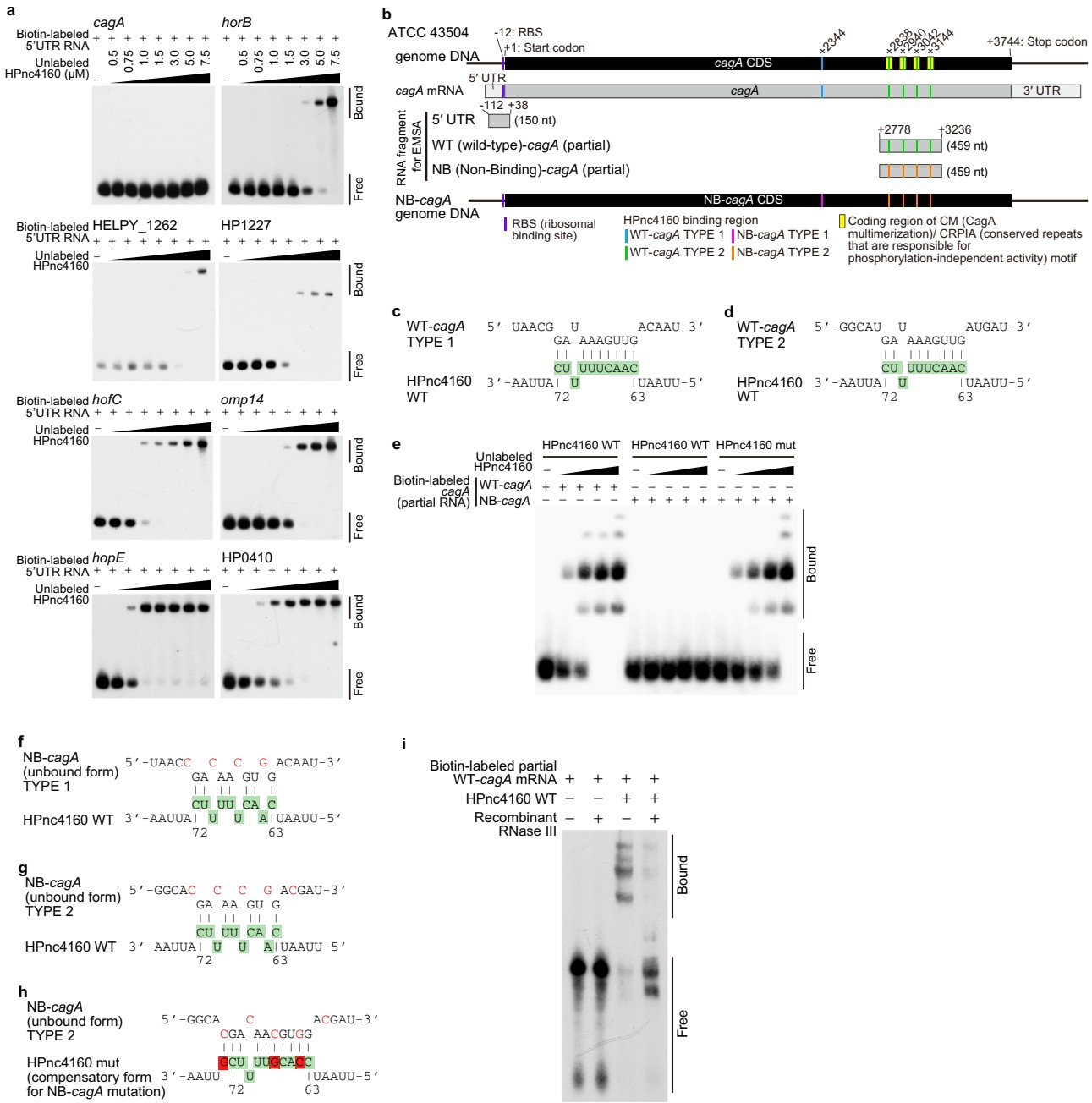

**Fig. 3 HPnc4160 binds to target mRNA. a** Electrophoretic mobility shift assay (EMSA) analysis of HPnc4160 binding to the 5′UTR region of each candidate mRNA. Data are representative of two independent experiments. **b** Schematic of CagA motifs, HPnc4160 binding regions, and HPnc4160-non-binding *cagA* (NB-*cagA*). **c, d** Schematic of predicted HPnc4160 binding sites in the corresponding CDS sequence of *cagA* TYPE 1 (**c**) and TYPE 2 (**d**). Upper sequences indicate target *cagA* mRNA sequences; lower sequences indicate the HPnc4160 sequence with base numbers. Colored sequences correspond to the loop structures indicated in Supplementary Fig. 4a. **e** EMSA analysis of HPnc4160 WT or HPnc4160 mut (a compensatory form of NB-*cagA*) bindings to RNA of partial *cagA* WT or NB-*cagA*. Data are representative of two independent experiments. **f–h** Schematic of CDS sequence of the NB-*cagA* of TYPE 1 (**f**) and TYPE 2 (**g**) and HPnc4160 mut (**h**). Upper sequences indicate target *cagA* mRNA sequences; lower sequences indicate the HPnc4160 sequence with base numbers. Green-colored sequences correspond to the loop structures of HPnc4160 indicated in Supplementary Fig. 4a. Mutated nucleotides in the *cagA* mRNA sequence are shown in red. Red-colored sequence correspond to mutated nucleotides in the *hpnc4160* sequence (**h**). **i** RNase protection assay with HPnc4160, *cagA* mRNA, and recombinant RNase III. Data are representative of two independent experiments. Source data are provided as a Source Data file.

IL-8 in AGS cells infected with the T-repeat mutated strains (Supplementary Fig. 5j).

Next, to understand the significance of the HPnc4160 control mechanism in bacterial adaptation to the host, mice were orally inoculated with either wild-type, Δ*hpnc4160-hpnc4170*, Δ*hpnc4160-hpnc4170*/pHel2-*hpnc4160*, and NB-*cagA*, and the

bacterial loads in the stomach were analyzed. At three days post-infection, a timepoint prior to T-repeat length extension (Supplementary Fig. 2e), levels of bacterial colonization and *Cxcl2* mRNA in the stomach were found to be significantly increased for the Δ*hpnc4160-hpnc4170* strain compared to wild-type, whereas those of the Δ*hpnc4160-hpnc4170*/pHel2-*hpnc4160*

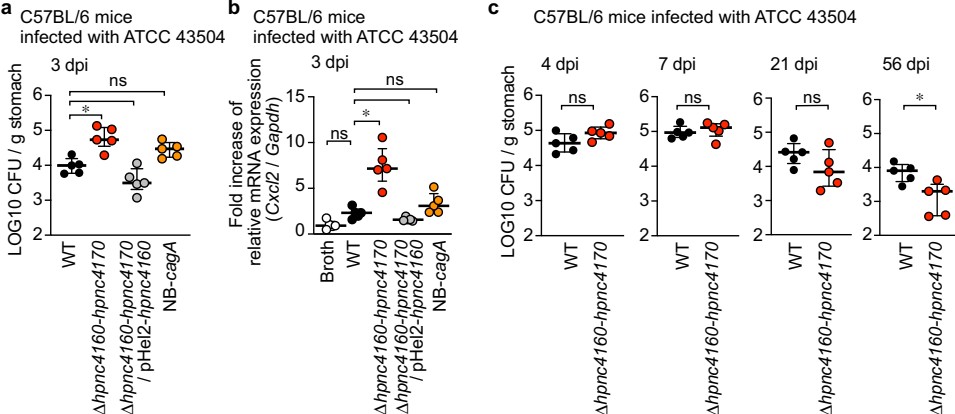

**Fig. 4 HPnc4160 regulates bacterial-host adaptation and pathogenesis. a, b** Three days after infection, quantitative culture assays (**a**) and RT-PCR (**b**) were performed on mouse gastric specimens. **c** Effect of HPnc4160 deletion of *H. pylori* on bacterial load in mouse stomach at indicated days of post infection (dpi). Statistical significance determined by non-parametric Dunn's multiple comparison test (two-sided) (**a, b**) or two-tailed Mann–Whitney test (**c**). *P = 0.0485 (**a**); *P = 0.0309 (**b**); *P = 0.0159 (**c**); ns, not significant. Source data are provided as a Source Data file.

strain were equivalent to wild-type (Fig. 4a, b). Notably, both colonization levels and *Cxcl2* mRNA levels were slightly higher following NB-*cagA* infection than following wild-type infection. This result indicates that factors controlled by HPnc4160, other than CagA, are important for bacterial adaptation and host gastritis development (Fig. 4a, b). Importantly, the bacterial colonization levels at 8 weeks post infection was significantly decreased in the Δ*hpnc4160-hpnc4170*-infected animals, as compared to the wild-type infection (Fig. 4c), indicating that HPnc4160 is beneficial for long-term colonization of *H. pylori*.

**Effects of T-repeats length diversity on adaptation.** Since *H. pylori* is known to be transmitted from human to human, we conducted reinfection experiments using isolates from mice with different T-repeats (T16 in Hp6 and T18 in Hp8) with equivalent in vitro growth rates (Supplementary Fig. 6a). Similar to the wild-type strain (Fig. 4a, c), both strains used for reinfection showed almost the same number of colonized bacteria in the stomachs from 3 days to 8 weeks after infection (Supplementary Fig. 6b). Interestingly, the strains used for reinfection displayed increased repeat lengths with increases in infection period, similar to the wild-type strain (Supplementary Figs. 2e and 6c). Furthermore, the strains with longer T-repeats (T16 and T18) were more likely to acquire the T-repeat elongation and adapt to the mice stomach than the strains with shorter T-repeats (T1 and T4) (Fig. 5a, b). These data indicate that repetitive sequence length variation of the T-repeat is advantageous for *H. pylori* persistent infection.

**T-repeat length and HPnc4160 expression in clinical isolates.** Because one of the HPnc4160 target factors was CagA, which is known as the strongest risk factor for *H. pylori*-related gastric cancer, we examined the expression of HPnc4160 target genes in clinical isolates from non-cancer patients and patients with cancer. Sequence analysis of clinical isolates showed that T-repeat regions were longer in strains derived from patients with cancer than in those derived from non-cancer patient (Fig. 6a, Supplementary Fig. 7, Supplementary Data 7). As shown in Fig. 6b, isolates from patients with cancer had lower levels of *hpnc4160* but increased expression of six factors controlled by HPnc4160 (*cagA*, *horB*, *hopE*, *omp14*, *hofC*, and HP0410) compared to isolates from non-stomach cancer patients (Fig. 6b, Supplementary Fig. 7b, c). The relevance of HPnc4160 and *H. pylori*-related gastric cancer risk was also assessed by in vitro infection experiments. AGS cells infected with *H. pylori* expressing low

levels of HPnc4160 displayed increased expression of *CDX2*, an indicator of intestinal epithelialization in the precancerous state (Supplementary Fig. 7d). In summary, our mutational analysis studies revealed a functional small RNA that regulates the pathogenicity of *H. pylori* and may have implications for gastric cancer development.

## Discussion
We have demonstrated that mRNA expression of CagA and OMPs is suppressed by HPnc4160 at the onset of *H. pylori* infection and that over the course of infection, thymidine repeats are inserted into the upstream region of *hpnc4160*, decreasing HPnc4160 expression, resulting in increased target mRNA expression; these factors contribute to bacterial adaptation to the host environment and potentiate gastritis and gastric oncogenesis (Fig. 7). Control of gene expression by varying the number of repeat sequences is a known control mechanism employed by *H. pylori*[23]. The present study suggests that repeat sequences of the *H. pylori* genome are important, not only as an ON/OFF mechanism for protein expression, including that of cell adhesion factors SabA and BabA[24], but also in sRNA expression. Since T-repeat is reported to affect the local DNA structure for RNA polymerase binding, by shifting the axial distance between the core promoter and upstream promoter elements, we speculate that steric distance between the promoter and the HPnc4160 transcription initiation site fluctuates as the T-repeat length increases or decreases, thereby modulating HPnc4160 expression[12,25]. As variations in the T-repeat length upstream of *hpnc4160* in *H. pylori* were observed in vivo (Figs. 1c and 5b; Supplementary Figs. 2c–f and 6c), through slipped strand mispairings[26], the *H. pylori* population may have become genetically heterogeneous during the course of infection, ensuring that a bacterium suitable for host colonization was selected and propagated. Importantly, the repetitive sequence length variation is advantageous for prolonged infection of *H. pylori* (Fig. 5a). In this study, we primarily used the *H. pylori* strain ATCC 43504, a clinical isolate originating from the human antrum. The T-repeat length did not fluctuate during in vitro subculture of ATCC 43504, but was found to increase upon isolation from infected rodents and ranged from 14–19 of T-repeat (Fig. 1c; Supplementary Fig. 2c–g), indicating that slipped strand mispairings were induced by stress conditions exerted by the host.

The HPnc4160-binding sequence appears five times in the ATCC 43504 *cagA* CDS within the CM/CRPIA motifs in the CagA C-terminal region, which is known to bind with host signal

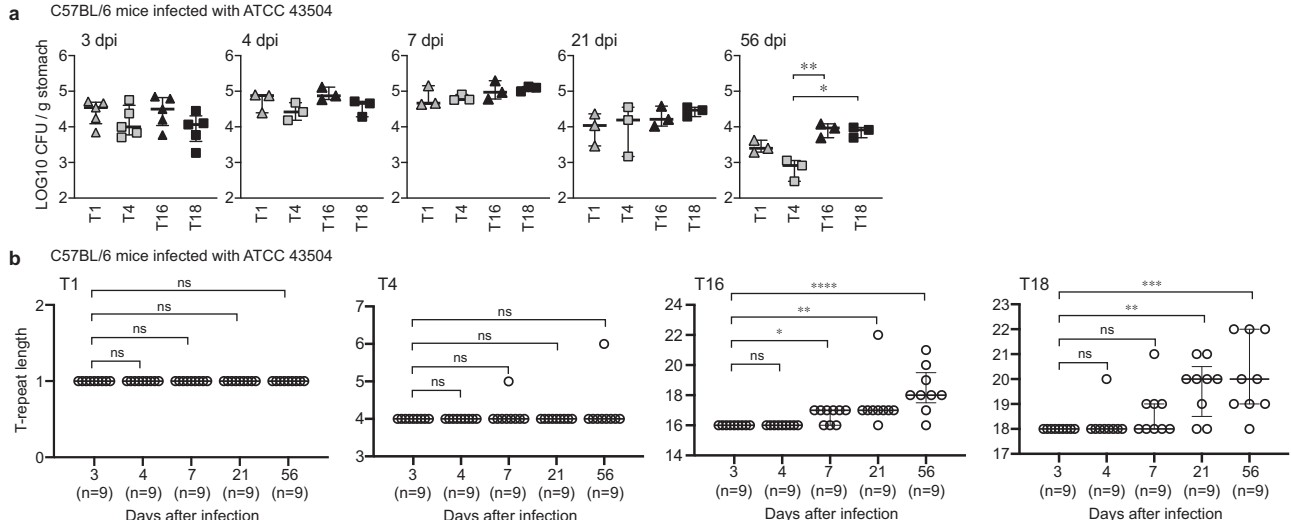

**Fig. 5 Time-dependent changes in bacterial-host adaptation and the T-repeat length. a, b** Time-dependent changes in the bacterial load in the infected mouse stomach (**a**) and length of the T-repeat region (**b**) in T1, T4, T16 and T18 strains recovered from mice. Data are presented as the median with interquartile range. Statistical significance determined by uncollected Dunn's multiple comparison test (two-sided). *$P = 0.0174$, **$P = 0.0066$ (**a**); *$P = 0.0294$, **$P = 0.0014$, ****$P < 0.0001$ (**b**, T16); **$P = 0.0012$, ***$P = 0.0002$ (**b**, T18); ns, not significant. Source data are provided as a Source Data file.

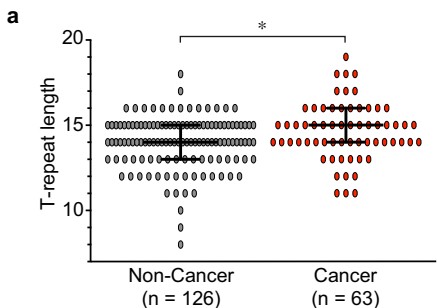

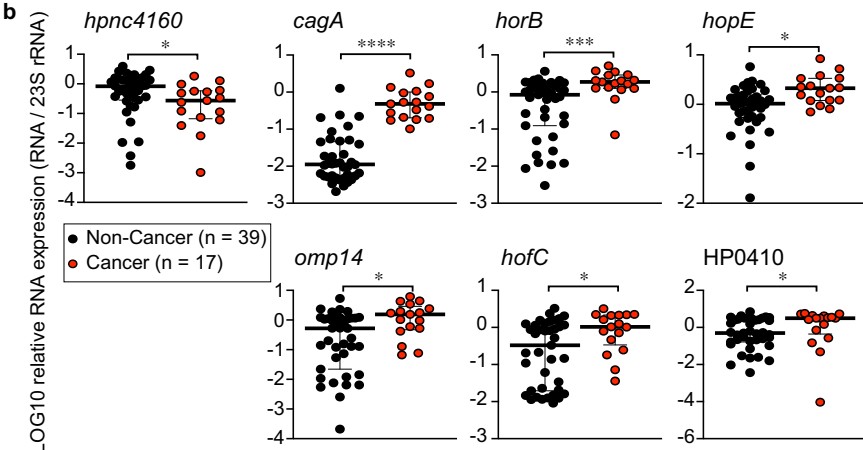

**Fig. 6 T-repeat length and HPnc4160 expression in clinical isolates. a, b** T-repeat length upstream of HPnc4160 (**a**) and expression of indicated mRNAs (**b**) in clinical isolates from non-cancer (non-cancer) and patients with cancer (cancer). Data are presented as means with 95% confidence interval (**a**) or medians with interquartile range (**b**). Statistical significance determined by two-tailed Mann–Whitney test (**a, b**). *$P = 0.0157$ (**a**); *$P = 0.0215$ (hpnc4160), ****$P < 0.0001$ (cagA), ***$P = 0.0004$ (horB), *$P = 0.0025$ (hopE), *$P = 0.0110$ (omp14), *$P = 0.0122$ (hofC), *$P = 0.0205$ (HP0410) (**b**). Source data are provided as a Source Data file.

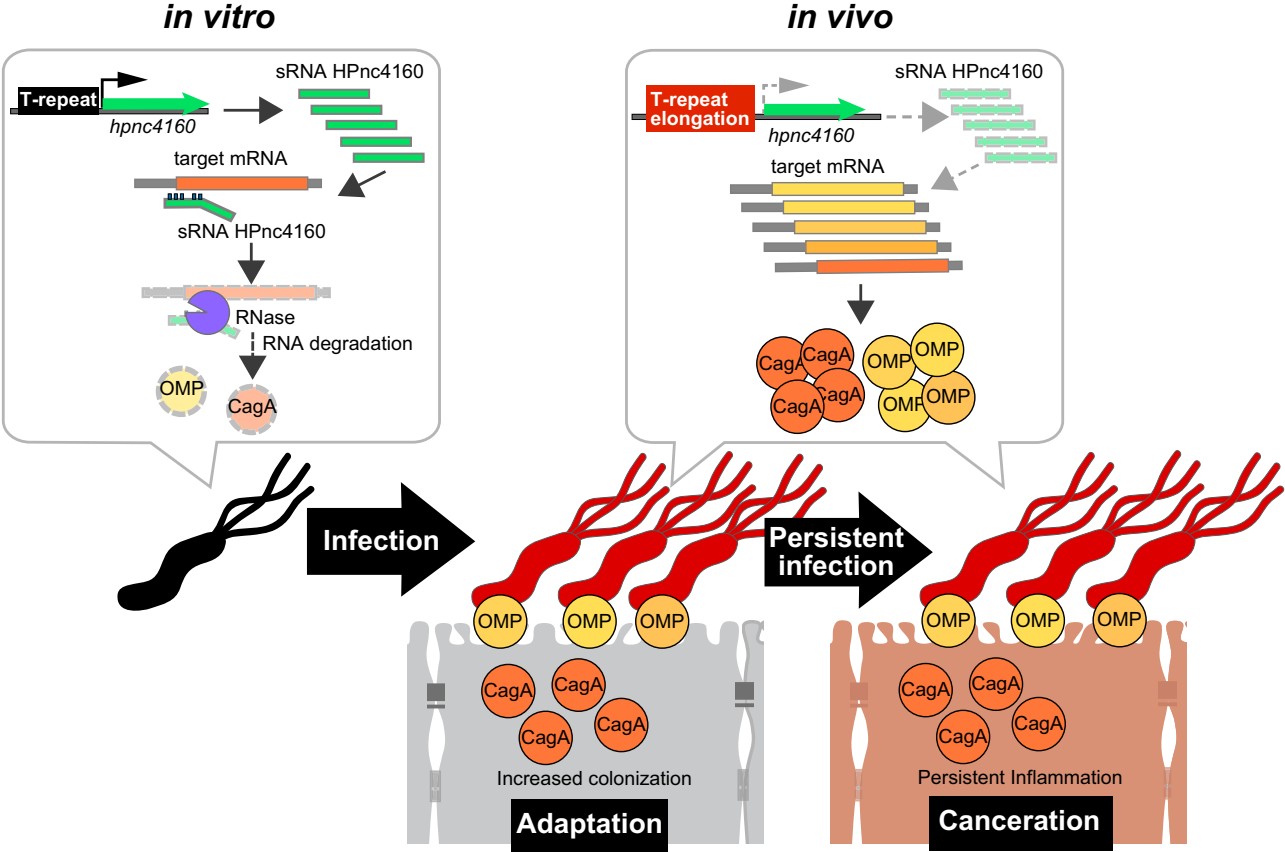

**Fig. 7 Infection-induced silencing of HPnc4160 upregulates target gene expression and promotes bacterial-host adaptation and, potentially, oncogenesis during chronic *H. pylori* infection.** *H. pylori* infection in vivo leads to elongation of T-repeats upstream of the HPnc4160 sRNA coding region, which results in decreased expression of HPnc4160 sRNA. Gene silencing of HPnc4160 results in increased target gene expression, coding for OMPs and CagA. As a result, bacterial colonization and CagA translocation into the attached host cells increase, promoting bacterial adaptation to the host and, possibly, oncogenesis.

proteins[27,28] (Fig. 3b). Generally, regulation of mRNA expression by sRNA is induced by binding at one site. Multiple binding sequences in the *cagA* CDS may facilitate efficient reduction of *cagA* transcript levels in response to HPnc4160. While the present manuscript was under revision, Eisenbart et al. published a paper describing that the nickel-responsive transcriptional regulator NikR regulates the expression of HPnc4160 (which the authors of that study renamed as 'NikS'), and HPnc4160 represses multiple major virulence genes, including *cagA* and *vacA* in vitro[29]. They showed that HPnc4160 binds to the 5′UTR of the *cagA* mRNA of strain G27. Although we could not observe HPnc4160 binding to the 5′UTR region of the ATCC 43504 *cagA* (Fig. 3a), they also identified a group of HPnc4160-regulating pathogenic factors that overlapped with those we found. Therefore, their report is in line with our finding that the sRNA HPnc4160 is a master regulator of *Helicobacter* pathogenicity.

All identified HPnc4160 targets encode large proteins that require greater energy expenditure for production (Fig. 2a–b, Supplementary Data 5 and 6). As the growth of bacterial cells in vitro may not require these pathogenic factors, *hpnc4160* expression may be higher whereas its targets are suppressed in vitro. Upon entering the host stomach, *H. pylori* may activate a mechanism that decreases HPnc4160 production, thereby simultaneously increasing expression of OMPs and CagA, allowing the bacteria to efficiently adapt to environmental changes and colonize the gastric mucosa.

Because CagA is a carcinogenic protein related to gastric cancer, an increase in CagA levels achieved by down-regulation of

HPnc4160 may correlate with an increased risk of gastric cancer. Additionally, we demonstrated that several OMPs are also regulated by HPnc4160. These targets are likely involved in host adaptation, given that the HPnc4160 knockout strains display increased bacterial colonization and inflammatory cytokine expression, phenotypes not solely dependent on CagA production (Fig. 4a, b). Collectively, we find HPnc4160 behaves as a master regulator of *H. pylori* host adaptation and may serve to potentiate gastric carcinogenesis.

## Methods

**Data reporting**. No statistical methods were used to predetermine sample size, the experiments were not randomized, and the investigators were not blinded to allocation during experiments and outcome assessment.

**Strains and culture conditions**. The isogenic mutants ΔcagA and ΔvirB7 of *Helicobacter pylori* strain ATCC 43504 were constructed in our laboratory[11]. The strain PMSS1 was generously provided by Dr, Manuel R. Amieva (Stanford University)[30]. *H. pylori* was cultured on Trypticase soy agar with 5% (v/v) sheep's blood (Thermo Fisher Scientific, Waltham, MA, USA) for 2 days at 37 °C in microaerobic conditions. Bacterial colonies were suspended in Brucella broth (Thermo Fisher Scientific) supplemented with 5% (v/v) inactivated fetal bovine serum (FBS; Thermo Fisher Scientific), adjusted to an optical density (OD) of 0.05 at 600 nm, and incubated for 15 h at 37 °C with gentle agitation under microaerobic conditions.

The AGS human gastric epithelial cell line (ATCC CRL-1739) was maintained in Dulbecco's modified eagle medium (DMEM)/F-12 (Thermo Fisher Scientific) containing 10% (v/v) FBS. AGS cells were seeded in six-well plates and grown to ~80% confluence to be used for western blot analysis. For immunofluorescence microscopy, cells were seeded in six-well plates with cover glass and grown to ~80% confluence[11].

**Antibodies and immunohistochemical reagents**. The anti-Tyr(P)-CagA (anti-pY-CagA; for Western blot, 1:100; for immunofluorescence microscopy, 1:10), anti-UreA (for Western blot, 1:2000), and anti-VirB7 (for Western blot, 1:200) polyclonal antibodies were made from the antiserum of rabbits immunized with recombinant proteins or peptides[11]. Anti-CagA polyclonal antibodies (for Western blot, 1:1000) were purchased from AUSTRAL Biologicals (CA, USA), anti-actin monoclonal antibodies (for Western blot, 1:1000) were from MERCK (Darmstadt, Germany), horseradish peroxidase (HRP)-labeled anti-rabbit IgG (for Western blot, 1:100,000) and HRP-labeled anti-mouse IgG (for Western blot, 1:5000), and fluorescein isothiocyanate-labeled anti-rabbit IgG (for immunofluorescence microscopy, 1:100) was from Jackson ImmunoResearch Laboratories Inc. (PA, USA). DAPI (for staining, 300 nM) was from Sigma-Aldrich (MD, USA), and rhodamine-phalloidin (for staining, 1:100) was from Thermo Fisher Scientific (MA, USA).

**Animal infection**. *H. pylori* infection of rodents was performed as follows[31,32]. Briefly, 6-week-old male MON/Jms/GbsSlc Mongolian gerbils were orally administered 200 μL Vancomycin (500 mg/L) 24 and 48 h before *H. pylori* inoculation. On the days of *H. pylori* inoculation, 300 μL 5% (w/v) sodium bicarbonate was orally administrated 10 min before bacterial inoculation. The gerbils were then intragastrically inoculated with an *H. pylori* culture containing $10^9$ colony forming units (CFU) for 2 consecutive days. C57BL/6 mice (SLC Japan, Tokyo, Japan) were intragastrically inoculated once with *H. pylori* culture of $10^9$ CFU. After the indicated date, the stomach of each infected animal was opened along the greater curvature. To quantitatively isolate *H. pylori*, the stomach was excised, weighed, and homogenized. Serial dilutions were plated on *H. pylori*-selective agar plates (Eiken Chemical Co.) and incubated under microaerophilic conditions at 37 °C for 4 days, after which the CFU were counted with a minimal detection limit of $1 \times 10^3$ CFU.

To isolate strains recovered from *H. pylori*-infected rodents, each colony on the *H. pylori*-selective agar plates was picked up by disposable inoculating loop and spread on Trypticase soy agar with 5% (v/v) sheep's blood, and incubated under microaerophilic conditions at 37 °C for two days. Then, colonies were suspended in Brucella broth supplemented with 5% (v/v) inactivated FBS, adjusted to an OD of 0.05 at 600 nm, and incubated for 15 h at 37 °C with gentle agitation under microaerobic conditions. The cultures were preserved with 50% (v/v) glycerol in −80 °C until use.

**For RNA isolation, the tissue was immediately frozen in liquid nitrogen**. All the animals were housed under controlled conditions with a 12 h light/dark cycle, 20-22 °C and 45 ± 5% humidity. Animal experiments were conducted in accordance with the University of Tokyo or Osaka University guidelines for the care and use of laboratory animals and were approved by the Ethics Committee for Animal Experiments at the University of Tokyo or Osaka University.

**Genomic DNA purification and sequencing**. For PCR templates, genomic DNA was purified using InstaGene Matrix (Bio-Rad Laboratories, CA, USA).

For whole-genome sequencing, genomic DNA was purified from mid-log phase cultures of strain ATCC 43504 using QIAGEN DNeasy (QIAGEN). A genomic DNA library for sequencing was prepared using the Nextera XT DNA Sample Preparation kit (Illumina, San Diego, CA, USA) and sequenced using the Illumina MiSeq (for isolates from gerbils) or HiSeq X (for isolates from mice) platform to generate 300-bp paired-end reads. Genome assembly, scaffolding, and gap-closing were performed using the Platanus assembler[33]. Gene identification and annotation were conducted by the Microbial Genome Annotation Pipeline (MiGAP).

DNA sequences mutated in >50% of the 40 strains recovered from Mongolian gerbils, or, in all of the 10 strains recovered from C57BL/6 mice are listed in Supplementary Data 2 and 4. mRNA expression analysis was performed as follows: for genes in which the mutation occurred in the CDS region, mRNA expression of the CDS was measured; for mutations in the intergenic region, mRNA expression of an adjacent gene in which the intergenic region could be a 5′UTR region was measured. HP1243 and HPG27_298, which started from the 3′ end of HP1243 with 33 nucleotides spaces, were regarded as a continuous gene since both genes are annotated as *babA*, and a ribosomal binding site (RBS) is assigned only at the upstream region of HP1243[34].

To confirm the number of T-repeats upstream of the HPnc4160 coding region, PCR was performed using primers designed to amplify around the region (Supplementary Data 8), and amplicons were purified by agarose gel electrophoresis and sequenced using an ABI3130xI DNA sequencer (Applied Biosystems, Foster City, CA, USA).

**In vitro passage experiment**. *H. pylori* ATCC 43504 was recovered from frozen stock and cultured on 5% (v/v) sheep's blood agar for 2 days at 37 °C in microaerobic conditions. Bacterial colonies were suspended in 3 tubes of Brucella broth supplemented with 5% (v/v) inactivated FBS. Each bacterial suspension was adjusted to an OD of 0.05 at 600 nm and incubated for 12 h at 37 °C with gentle agitation under microaerobic conditions. Subsequently, each fraction of the suspension was preserved by freezing in 50% (v/v) glycerol as the "Original" strains. Meanwhile, each bacterial suspension was sub-cultured by resuspending in Brucella broth supplemented with 5% (v/v) inactivated FBS adjusted to an OD of 0.05 at 600 nm, and incubated an additional 12 h at 37 °C with gentle agitation under microaerobic conditions. Sub-cultivation was repeated for 60 passages (30 days). Then, each cell suspension was preserved by freezing in 50% (v/v) glycerol as the "60-passaged" strains. The "Original" and "60-passaged" strains were recovered from frozen stock on 5% (v/v) sheep's blood agar by 2 days of incubation under microaerobic conditions. Then, colonies were suspended in Brucella broth supplemented with 5% (v/v) inactivated FBS and incubated 12 h at 37 °C with gentle agitation under microaerobic conditions. The bacterial cells were collected and subjected to genomic DNA purification.

**RT-PCR**. To prepare total RNA from *H. pylori*, liquid *H. pylori* cultures were agitated under microaerobic conditions at 37 °C overnight until the OD value at 600 nm reached 0.9.

Total RNA was extracted using ISOGEN (Nippon Gene, Tokyo, Japan), according to the manufacturer's instructions. The concentration of purified total RNA was analyzed using a NanoDrop Spectrophotometer (ThermoFisher Scientific, Wilmington, DE, USA). Total RNA was reverse transcribed into cDNA with a miScript II RT Kit (QIAGEN) according to the manufacturer's instructions[35]. mRNA expression was quantified and normalized to that of 23SrRNA (for *H. pylori*) or *Gapdh* (for mice and human) expression with THUNDERBIRD SYBR qPCR (TOYOBO) using the primer pairs described in Supplementary Data 8[36]. The results are expressed as the means ± SEM from triplicate strain experiments.

**Northern blot analysis**. 2 μg of total RNA in RNA loading dye (NEB) was separated by gel electrophoresis on 6% polyacrylamide/7 M urea gels in 1xTBE buffer for 3 h at 250 V and was electroblotted onto Hybond-XL nylon membrane (GE Healthcare) for 1 h at 50 V using Biometra Eco-Maxi system (Analytik-Jena). The membrane was crosslinked by 120 mJ/cm² UV light. Oligonucleotides JVO-2624 and JVO-0485 to detect HPnc4160 and 5 S rRNA, respectively[9], were 5′-end-labeled with [$^{32}$P]-γ-ATP by T4 polynucleotide kinase (Nippon Gene) and purified over G25 columns (GE Healthcare). After prehybridization in Rapid-Hyb buffer (Amersham), the [$^{32}$P]-labeled probe was hybridized at 42 °C overnight. Membrane was washed in three steps in 5x SSC/0.1% SDS, 1x SSC/0.1% SDS and 0.5x SSC/0.1% SDS buffers for 15 min at 42 °C. Signals were visualized on Typhoon FLA7000 scanner (GE Healthcare).

**Genetic manipulation**

*Construction of plasmids for producing gene-deficient mutants.* Isogenic gene null mutants derived from ATCC 43504 were constructed by insertional mutagenesis as follows[37]. Using the extracted *H. pylori* ATCC 43504 genome as a template, DNA fragments containing the 500 bp upstream region and the 500 bp downstream region of the target gene were amplified by PCR using the primers (CagA KO up XhoI, CagA KO up EcoRI, CagA KO down BamHICagA KO down NotI, HPnc4160/HPnc4170 KO up KpnI, HPnc4160/4170 KO up ClaI, HPnc4160/HPnc4170 KO down BamHI, HPnc4160/HPnc4170 KO down SacI) listed in Supplementary Data 8. The DNA fragments were introduced at both sides of *aphA3* (which confers kanamycin resistance) in pBluescript II SK ( + ) plasmids. The fragments from the resulting plasmids were introduced into *H. pylori* by electroporation.

*Construction of non-marker H. pylori mutants.* To construct non-marker *H. pylori* mutants, ATCC 43504 *flaA* and *cag1* promoters and terminators were cloned into pBluescript SK( + ) SmaI aphA3 SmaI, and *sacB* was cloned into the *EcoR*I site (pKSB plasmid).

Mid-log-phase (OD600 = 0.5–0.7) of *H. pylori* in 20 ml liquid culture were washed twice with ice-cold 10% glycerol and resuspended by 200 μL ice-cold 10% glycerol. One microgram of pKSB vector containing the target mutation and the bacterial cells were mixed at 4 °C and electroporated by a Micropulser (Bio-Rad) at the Ec2 (2.5 kV) setting. After 4 h of incubation at 37 °C in a microaerophilic condition, cells were plated on 5% sheep's blood agar TSAII plates containing 4 μg/mL kanamycin and incubated 2–3 days at 37 °C under microaerophilic conditions[38]. Four single colonies were seeded onto new 5% sheep's blood agar TSAII plates supplemented with 4 μg/mL kanamycin and incubated for an additional 2 days. Each colony was picked up and cultured in Brucella broth containing 5% FBS at 37 °C under microaerophilic conditions until *H. pylori* reached the mid-log phase. The medium (100 μL) was plated on 5% sheep's blood agar plates supplemented with 2.5% sucrose and cultured for 2 days. Each colony was then seeded onto a new 5% sheep's blood agar plate without antibiotics and incubated for 2 days. At the same time, colonies were also seeded onto a different agar plate with 4 μg/ml kanamycin to confirm that kanamycin resistance had been abolished. Surviving *H. pylori* were then transferred to liquid culture, and the genome sequence was confirmed by Sanger sequencing.

*Construction of point-mutated H. pylori.* The recombinant plasmids to establish various mutant T15, T16, T17, T18, and T19 *H. pylori* strains in the upstream region of *hpnc4160* were constructed by PCR using genomic *H. pylori* DNA from the strains isolated from gerbils after 8 weeks as a template (primers: pKSB-HPnc4160 Point mut ApaI and pKSB- HPnc4160 Point mut XhoI; listed in

Supplementary Data 8), then, the resulting DNA fragments were cloned into suicide pKSB plasmids. To construct the recombinant plasmids to establish various mutant *H. pylori* strains in the upstream region of *hpnc4160* (T0, T1, T2, T3, T4, T5, T6, T7, T8, T9, T10, T11, T12, T13, and T20), mutated DNA fragments were constructed using PrimeSTAR Mutagenesis Basal Kit (Takara, Shiga, Japan) (template: pKSB HPnc4160 T15 plasmid; primers: T0-repT_s and T0-repT_as, T1-repT_s and T1-repT_as and T3-repT_s and T3- repT_as, T4-repT_s and T4-repT_as, T5-repT_s and T5-repT_as, T6-repT_s and T6-repT_as, T7-repT_s and T7-repT_as, T8-repT_s and T8-repT_as, T9-repT_s and T9-repT_as, T10-repT_s and T10-repT_as, T11-repT_s and T11-repT_as, T12-repT_s and T12-repT_as, T13-repT_s and T13-repT_as, T20-repT_s and T20-repT_s; listed in Supplementary Table 8). *H. pylori* T0, T1, T2, T3, T4, T5, T6, T7, T8, T9, T10, T11, T12, T13, T15, T16, T17, T18, T19, and T20 mutants were established by independently introducing the pKSB-based plasmids into the *H. pylori* ATCC 43504 strain.

*Construction of NB-cagA-expressing H. pylori.* Based on the full-length cagA cDNA sequence of ATCC 43504, we designed a HPnc4160-unbound *cagA* gene sequence (NB-*cagA*; Supplementary Fig. 5f, g). The NB-*cagA* cDNA was artificially synthesized as 908 bp pEX-K4J2-cagA mutants (Eurofins, 99900008281-1). cDNA fragments containing the mutated *cagA* sequence were amplified (primers: pKSB-cagA-NB-ApaI, pKSB-cagA-NB-XhoI, listed in Supplementary Data 8), and cloned into suicide pKSB plasmids. The resulting plasmids were introduced to *H. pylori* ATCC 43504 to obtain NB-*cagA*-expressing *H. pylori*.

*Construction of hpnc4160 over-expressing H. pylori.* Plasmids for overexpression of *hpnc4160* in *H. pylori* were constructed by combining DNA fragments of *hpnc4160* regions amplified by PCR using the primers pHel2- HPnc4160-de-4170-hed-f XhoI and pHel2- HPnc4160/HPnc4170-de-4170-hed-r BamHI (Supplementary Data 8) and genome DNA of the ATCC 43504 strain as a template. The resulting DNA fragments included the upstream region of *hpnc4160*, excluding the 5′ region of the *hpnc4170* region. The DNA was cloned into a pHel2 shuttle vector[39] and introduced into *H. pylori* by electroporation.

**RNA-seq.** *H. pylori* were agitated under aerobic conditions and cultured at 37 °C overnight until the OD value at 600 nm reached 0.9. Total RNA from *H. pylori* was extracted using RNeasy (QIAGEN) following the manufacturer's instructions. The concentration of total RNA extracted was examined using a NanoDrop Spectrophotometer (ThermoFisher Scientific, Wilmington, DE, USA), according to the manufacturer's instructions.

Ten micrograms from each total RNA sample were treated with the MICROBExpress Bacterial mRNA Enrichment kit (Ambion, Grand Island, NY, USA) and RiboMinus™ Transcriptome Isolation Kit (Bacteria) (Invitrogen, Grand Island, NY, USA) following the manufacturer's instructions. Samples were resuspended in 15 μL RNase-free water. Bacterial mRNAs were chemically fragmented to 200-250 bp using 1 × fragmentation solution (Ambion, Grand Island, NY, USA) for 2.5 min at 94 °C. cDNA was generated according to the instructions given in a SuperScript Double-Stranded cDNA Synthesis Kit (Invitrogen, Grand Island, NY, USA). Briefly, each mRNA sample was mixed with 100 pmol random hexamers, incubated at 65 °C for 5 min, chilled on ice, mixed with 4 μL First-Strand Reaction Buffer (Invitrogen, Grand Island, NY, USA), 2 μL 0.1 M DTT, 1 μL 10 mM RNase-freed NTPmix, 1 μL SuperScript III reverse transcriptase (Invitrogen), and incubated at 50 °C for 1 h. To generate the second strand, the following Invitrogen reagents were added: 51.5 μL RNase-free water, 20 μL second-strand reaction buffer, 2.5 μL 10 mM RNase-free dNTP mix, 50 U *E. coli* DNA Polymerase, 5 U *E. coli* RNase H, and incubated at 16 °C for 2.5 h. The Illumina Paired-End Sample Prep kit was used for RNA-seq library creation according to the manufacturer's instructions as follows: fragmented cDNA was end-repaired, ligated to Illumina adaptors, and amplified by 18 cycles of PCR. Paired-end 150-bp reads were generated by high-throughput sequencing with an Illumina Hiseq 2500 Genome Analyzer. After removing the low-quality reads and adaptors, RNA-seq reads were aligned to the corresponding ATCC 43504 genome using Tophat 2.1.0[40], allowing for a maximum of two mismatches. If reads mapped to more than one location, only that with the highest score was kept. Reads mapping to rRNA and tRNA regions were removed from further analysis. After obtaining read numbers from every sample, edgeR with TMM normalization method was used to determine differentially expressed genes. Significantly differentially expressed genes (FDR value < 0.05 and at least two-fold change) were selected for further analysis[41].

**iTRAQ.** *H. pylori* ATCC 43504 strains of wild-type, Δ*hpnc4160-hpnc4170*, and Δ*hpnc4160-hpnc4170* / pHel2-*hpnc4160* were cultured in Brucella broth containing 5% fetal calf serum (FCS) to OD600 = 0.9. Each bacterial solution (1.5 mL) was centrifuged at 5000 ×*g* for 10 min at 4 °C. The resulting pellet was resuspended in wash buffer (1 M KCl, 15 mM Tris-HCl, pH 7.4), centrifuged again, and the supernatant was removed[42]. The pellet was resuspended in a Wash buffer containing 1 mM AEBSF (4- (2-Aminoethyl) benzenesulfonyl fluoride hydrochloride) and frozen at −80 °C. iTRAQ analysis was commissioned to Filgen Corporation. Data were analyzed using ProteinPilot software (SCIEX, USA). Total ProtScore is a measure of the total amount of evidence for a detected protein and calculated using

all of the peptides detected for the proteins. P values are measures of the certainty that the average ratio differs between samples.

**EMSA (electrophoretic mobility shift assay).** cDNA fragments of the entire small RNA *hpnc4160* region; 150-bp fragments of the 5′UTR regions [from 100 bases upstream of the ribosome binding region (RBS), to 50 bases downstream of the RBS] of *hp0410* gene, *hp0486* gene, *horB* gene, *hp0671* gene, *hopE* gene, *cagA* gene, *hp1227* gene, and *helpy_1262* gene; and 459 bp of the total cDNA containing the four *hpnc4160*-binding regions near the 3′tail of *cagA*, were amplified by PCR (primers: Small RNA HPnc4160 XhoI, Small RNA HPnc4160 EcoRI; HP0410 150 bp XhoI, HP0410 150 bp EcoRI; HELPY_0660 150 bp XhoI, HELPY_0660 150 bp EcoRI; HP0671 150 bp XhoI, HP0671 150 bp EcoRI; HP0486 150 bp XhoI, HP0486 150 bp EcoRI; HPSH_00635 150 bp XhoI, HPSH_00635 150 bp EcoRI; HPP12_0555 150 bp XhoI, HPP12_0555 150 bp EcoRI; HP1227 150 bp XhoI, HP1227 150 bp EcoRI; HELPY_1262 150 bp XhoI, HELPY_1262 150 bp EcoRI; CagA-B coding XhoI, CagA-B coding EcoRI; listed in Supplementary Data 8) and the ATCC 43504 genome as a template. The PCR products were cloned into the pBluescript SK ( + ) plasmid downstream of the T7 promoter region. pBluescript SK ( + ) plasmid containing HPnc4160 mut (compensatory form for NB-*cagA* mutation) was generated according to the instruction of PrimeSTAR Mutagenesis Basal Kit (Takara) using pBluescript SK ( + ) *hpnc4160* as a template (primers: HPnc4160_pointmut_primer_f, HPnc4160_pointmut_primer_r; listed in Supplementary Data 8). The NB-*cagA* mutant RNA used in the gel shift assay was amplified with a T7 promoter by PCR (T7 promoter CagA-NB EMSA PCR s, T7 promoter CagA-NB EMSA PCR as) and synthesized pEX-K4J2-CagA mutant (Eurofins, 99900008281-1) *cagA* as a template. *cagA* mutant RNA was prepared in the same manner except without mutations in the four HPnc4160-binding regions. RNA was transcribed from a DNA fragment using an in vitro Transcription T7 kit (Takara)[43].

Gel shift assays were performed using 0.04 pmol 3′-biotin-tagged mRNA with increasing amounts of purified small RNA HPnc4160 in 20 μL reactions. Briefly, RNA was denatured (10 min, 80 °C) and cooled for 5 min on ice. Yeast tRNA (1 μg) (ThermoFisher SCIENTIFIC) was added to the labeled RNA, and binding buffer (10 mM HEPES pH 7.3, 1 mM MgCl₂, 20 mM KCl, 5% glycerol) was added to a final volume of 10 μL[44]. Then, 10 μL labeled mRNA was added to HPnc4160. The mixtures were incubated at room temperature for 20 min. Then the samples were mixed with 5 μL native loading buffer before loading on a pre-cooled native 6% poly-acrylamide (PAA), 0.5× TBE gel. Gels were run in 0.5× TBE buffer at 30 mA per gel for 2 h[45,46].

**Cleavage assays.** The cDNA of 720 bps of *H. pylori rnase* III was amplified by PCR (primers: pGEX-6P-1 RNaseIII XhoI-f, pGEX-6P-1 RNaseIII NotI-r, listed in Supplementary Data 8; and template: genome DNA from ATCC 43504 strain). The cDNA was cloned into a pGEX6P-1 vector (GE). *E. coli* BL21 transformed with the plasmids were subjected to shaking culture in LB broth containing 100 μg/mL ampicillin at 37 °C with constant shaking at 200 r.p.m. Protein expression was induced with IPTG at a final concentration of 0.1 mM, at 4 °C, for 4 h. The bacteria were collected by centrifugation, and pellets were subjected to GST-fusion protein purification using Glutathione Sepharose 4B (GE) according to the manufacture's instruction. RNase III protein was excised by PreScission Protease according to the manufacturer's instructions. The purified protein derived from 6.7 mL of bacterial culture was developed by SDS-PAGE, and the gel was stained with Coomassie Brilliant Blue to confirm that no contaminants were observed in the final product. The protein concentration was determined by absorbance at 280 nm[44].

Nuclease assays using RNase III were performed using purified *H. pylori* recombinant RNase III. The gel shift assay protocol described above was followed, except that an RNase III-specific buffer (25 mM Tris pH 7.5, 50 mM NaCl, 50 mM KCl, 10 mM MgCl₂, 1 mM DTT) was used instead of Binding Buffer. 3′-biotin-tagged partial *cagA* mRNA was incubated on ice with 5 μM of small RNA HPnc4160 for 20 min. RNase III was then added at a final concentration of 300 nM, and the reactions were incubated for 1 min at 37 °C. The samples were mixed with 5 μL native loading buffer before loading on a pre-cooled native 6% PAA, 0.5x TBE gel[47].

**ELISA.** AGS cells were co-incubated with *H. pylori* at a multiplicity of infection (MOI) of 100 for 12, 24 and 36 h at 37 °C in a 5% CO₂ environment in 24 well plates. The supernatants were collected and stored at -30 °C. Enzyme-linked immunosorbent assays (ELISAs) for human IL-8 were performed using the Human IL-8 ELISA Kit (ThermoFisher SCIENTIFIC) according to the manufacturer's instructions. The results are expressed as the means ± SEM from triplicate experiments.

**Immunofluorescence microscopy.** AGS cells were infected with *H. pylori* at an MOI of 100 for 6 h at 37 °C in a 5% CO₂ environment. The cells were fixed with 4% (w/v) paraformaldehyde-PBS at room temperature for 10 min. The cells were then washed with TBS 3 times, and blocked with Saponin buffer [10% (v/v) Blocking One (Nakalai, Japan) containing 0.2% (w/v) saponin] at 4 °C for 60 min. Antibodies and fluorescent stains used for staining were DAPI, rhodamine-phalloidin (Thermo Fisher SCIENTIFIC, MA, USA), and anti-pY-CagA[11,28]. Confocal laser

scanning microscopy (CLSM) image acquisition was performed using a Zeiss LSM 800 confocal laser scanning microscope with ZEN 2.3 software (Carl Zeiss, Jena, Germany).

**Reporting summary**. Further information on research design is available in the Nature Research Reporting Summary linked to this article.

## Data availability

The raw read sequences and assembled scaffold sequences of *H. pylori* strains recovered from rodents or clinical isolates used in this study are available in the DDBJ/EMBL/Genbank databases under Bioproject accession numbers SAMD00178897 to SAMD00178935, SAMD00179460, SAMD00178937 and SAMD00204457 to SAMD00204466. Sequence data of the clinical isolates used in this study are available in the DDBJ/EMBL/Genbank databases with the accession codes listed in Supplementary Data 7. The authors declare that all other data supporting the findings of this study are available within the paper and its supplementary files. Source data are provided with this paper.

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

## Acknowledgements

The authors would like to thank Manuel Amieva for providing the *H. pylori* strains. We acknowledge Keisuke Katsura and Junko Akada for their support. We would like to thank the members of the Division of Bacteriology, Department of Infectious Diseases

Control, International Research Center for Infectious Diseases, The Institute of Medical Science, The University of Tokyo, and the members of the Department of Infection Microbiology, Research Institute for Microbial Diseases, Osaka University. This work was supported in part by Grant-in-Aid for Scientific Research from the Ministry of Education, Culture, Sports, Science, and Technology of Japan [17K19551, 18K07127, 19K22704 (to H.M.), 16K07083 (to T.S.), 17K14974 (to M.T.), 20K16244 (to R.K.-D.)], the Naito Foundation (to H.M.), the Tokyo Biochemical Research Foundation (to H.M. and P.S.), the Kao Foundation for Arts and Sciences (to K.K.), and the Smoking Research Foundation (to H.M.).

## Author contributions

R.K.-D., K.K. and H.M. conceived the study, designed experiments, and wrote the manuscript; M.M. performed Northern blot analysis; R.O. performed part of the mouse experiments; Z.B. analyzed gene-expression data sets; Y.O., R.Y., K.N. and T.H. analyzed genome sequences; T.V.P. and Y.Y. provided clinical isolates and analyzed genome sequences; T.Sa., T.O., T.I., E.K., S.H., M.T., A.S., P.S., H.A., and T.Su. provided technical assistance with experiments; T.T.B., L.T.N., K.V.V., and D.Q.D.H. provided clinical isolates; H.M. supervised the study.

## Competing interests

The authors declare no competing interests.
