## [Peer Review File · Nature Communications]

REVIEWERS' COMMENTS

Reviewer #1 (Remarks to the Author):

The authors have comprehensively addressed my original concerns with new experimentation.

Reviewer #2 (Remarks to the Author):

The authors have made a commendable effort to address the reviewers' comments. The study by Kinoshita-Daitoku et al., which examines the role of the small RNA in host adaptation and carcinogenesis, is very complementary to a study of the same sRNA by Eisenbart et al., which is more focused on the in vitro characterization of the small RNA. However, I think it is essential that Kinoshita-Daitoku et al. adopt the name, NikS, already published by Eisenbart et al. in *Mol. Cell*, especially since the NikS nomenclature follows the convention of naming small RNAs based on information about their expression. There is no need to confuse the field with multiple names for the same RNA.

Reviewer #4 (Remarks to the Author):

The authors have addressed the concerns and queries associated with the initial submission in a thoughtful and meaningful manner. This manuscript represents a strong advance in the field of *Helicobacter pylori* biology and disease.

Response to Referees letter

We would like to thank each of the three reviewers for their careful and thorough reading of our manuscript and their positive response to the work.

The following is a point-by-point response to the comments.

REVIEWERS' COMMENTS

Reviewer #1 (Remarks to the Author):

The authors have comprehensively addressed my original concerns with new experimentation.

Thank you for your approval.

Reviewer #2 (Remarks to the Author):

The authors have made a commendable effort to address the reviewers' comments. The study by Kinoshita-Daitoku et al., which examines the role of the small RNA in host adaptation and carcinogenesis, is very complementary to a study of the same sRNA by Eisenbart et al., which is more focused on the in vitro characterization of the small RNA. However, I think it is essential that Kinoshita-Daitoku et al. adopt the name, NikS, already published by Eisenbart et al. in Mol. Cell, especially since the NikS nomenclature follows the convention of naming small RNAs based on information about their expression. There is no need to confuse the field with multiple names for the same RNA.

Thank you for your suggestion. We would totally agree with your comment of "There is no need to confuse the field with multiple names for the same RNA". Therefore, we have removed an unnecessary new name 'MrhP' and used the original name 'HPnc4160' throughout the paper.

Reviewer #4 (Remarks to the Author):

The authors have addressed the concerns and queries associated with the initial submission in a thoughtful and meaningful manner. This manuscript represents a strong advance in the field of *Helicobacter pylori* biology and disease.

Thank you for your compliment.